# Negative linkage disequilibrium between amino acid changing variants reveals interference among deleterious mutations in the human genome

Jesse A. Garcia[1], Kirk E. Lohmueller[1,2,3]*

1 Interdepartmental Program in Bioinformatics, University of California, Los Angeles, California, United States of America, 2 Department of Ecology and Evolutionary Biology, University of California, Los Angeles, California, United States of America, 3 Department of Human Genetics, David Geffen School of Medicine, University of California, Los Angeles, California, United States of America

* klohmueller@ucla.edu

**Data Availability Statement:** High-coverage sequencing data is available from the New York Genome Center (http://ftp.1000genomes.ebi.ac.uk/vol1/ftp/data_collections/1000G_2504_high_

## Abstract

Evolutionary forces like Hill-Robertson interference and negative epistasis can lead to deleterious mutations being found on distinct haplotypes. However, the extent to which these forces depend on the selection and dominance coefficients of deleterious mutations and shape genome-wide patterns of linkage disequilibrium (LD) in natural populations with complex demographic histories has not been tested. In this study, we first used forward-in-time simulations to predict how negative selection impacts LD. Under models where deleterious mutations have additive effects on fitness, deleterious variants less than 10 kb apart tend to be carried on different haplotypes relative to pairs of synonymous SNPs. In contrast, for recessive mutations, there is no consistent ordering of how selection coefficients affect LD decay, due to the complex interplay of different evolutionary effects. We then examined empirical data of modern humans from the 1000 Genomes Project. LD between derived alleles at nonsynonymous SNPs is lower compared to pairs of derived synonymous variants, suggesting that nonsynonymous derived alleles tend to occur on different haplotypes more than synonymous variants. This result holds when controlling for potential confounding factors by matching SNPs for frequency in the sample (allele count), physical distance, magnitude of background selection, and genetic distance between pairs of variants. Lastly, we introduce a new statistic $H_R^{(j)}$ which allows us to detect interference using unphased genotypes. Application of this approach to high-coverage human genome sequences confirms our finding that nonsynonymous derived alleles tend to be located on different haplotypes more often than are synonymous derived alleles. Our findings suggest that interference may play a pervasive role in shaping patterns of LD between deleterious variants in the human genome, and consequently influences genome-wide patterns of LD.

coverage/working/20190425_NYGC_GATK/). The 1000 Genomes strict mask is available from http:// ftp.1000genomes.ebi.ac.uk/vol1/ftp/data_ collections/1000_genomes_project/working/ 20160622_genome_mask_GRCh38/StrictMask/. All scripts necessary for reproducing the figures are available at https://github.com/JesseGarcia562/ garcia_and_lohmueller_2020.

**Funding:** This work was supported by the National Institutes of Health (https://www.nih.gov/) grant R35GM119856 to KEL and a Gates Millennium Scholars Fellowship (gmsp.org) to JAG. The funders had no role in study design, data collection and analysis, decision to publish, or preparation of the manuscript.

**Competing interests:** The authors have declared that no competing interests exist.

## Author summary

Many mutations in genomes are deleterious, decreasing fitness in carriers. Popular methods to quantify deleterious mutations model mutations independently while ignoring the correlations between nearby variants. Theory predicts that a deleterious mutation can influence the frequency change of variants located nearby along the genome. Here we use simulations under population genetic models with parameters relevant to humans to show that pairs of deleterious mutations located near each other in the genome tend to have different correlations between them as compared to pairs of neutrally evolving SNPs. Specifically, if an individual carries the deleterious allele at one variant, that individual is less likely to carry the deleterious allele at a nearby variant. We then searched for these patterns in both low and high-coverage human genetic variation datasets from multiple populations. We found that pairs of deleterious alleles tend to be found in different individuals more frequently than are pairs of neutrally evolving variants at the same frequency, even after controlling for confounding factors. Our results suggest that the interference between deleterious alleles is common across the human genome, which has implications for inferring demographic history, natural selection, and associating variants with complex traits.

## Introduction

The non-random association of alleles at different loci is often referred to as linkage disequilibrium (LD). The magnitude of LD between two single nucleotide polymorphisms (SNPs) is shaped by both population processes, such as demographic history, and intrinsic cellular factors, like recombination and gene conversion [1–4]. If two variants are in linkage disequilibrium, they can either be in positive LD or negative LD. Positive LD occurs when derived alleles appear on the same haplotype more often than expected under independence (i.e. linkage equilibrium) and negative LD occurs when derived alleles appear on the same haplotype less often than expected under linkage equilibrium. Previous studies have used LD to estimate demographic parameters of populations such as the historical effective population size ($N_e$) and divergence times between populations [2,5,6]. Additionally, methods have been developed to estimate recombination rates from patterns of LD [7–11]. Most of the previous work on modeling patterns of LD has relied on assumptions about selective neutrality among markers, though some work quantified the effects of positive selection on LD [12–16].

Negative selection can influence patterns of LD in two ways: negative synergistic epistasis and Hill-Robertson Interference (HRI). Negative synergistic epistasis occurs when haplotypes carrying multiple deleterious mutations are less fit than predicted by their marginal fitness [17]. This leads to negative selection efficiently removing haplotypes containing multiple deleterious alleles. The remaining deleterious alleles are more likely to segregate on distinct haplotypes compared to neutral mutations, leading to negative LD [18]. Further, as individuals carrying many deleterious mutations are efficiently removed from the population, this synergistic epistasis leads to the distribution of deleterious mutations per genome being underdispersed [19,20]. Sohail *et al.* showed that rare loss-of-function alleles are underdispersed in human and *Drosophila* genomes [18], suggesting that polymorphisms currently segregating in human and *Drosophila* populations are not only experiencing negative selection, but are also non-independently affecting fitness.

A second way in which negative selection can impact LD is through Hill-Robertson interference (HRI) [21]. This scenario occurs when deleterious alleles are not efficiently removed

from the population by negative selection and can increase in frequency due to drift. Here, one deleterious variant inhibits or accelerates the removal of a cosegregating deleterious variant. [22]. When LD is positive, haplotypes carrying multiple deleterious variants are more effectively removed from this population than haplotypes containing only one deleterious variant because such haplotypes have the lowest fitness. In other words, positive LD between deleterious variants increases the rate at which the population responds to selection as the variance in fitness among individuals is greatest [22]. If, however, deleterious mutations occur on different haplotypes in a population, then the variance in fitness across haplotypes is reduced, as they all carry a similar number of deleterious alleles [23,24]. Consequently, selection becomes less efficient at removing haplotypes carrying one deleterious allele. In finite populations not in mutation-selection balance and experiencing drift, interference predicts pairs of deleterious SNPs will exhibit negative LD (relative to pairs of variants not under selection), especially when they are separated by small physical and genetic distances [25–27]. In our study, an excess of negative LD means the same thing as seeing fewer pairs of variants in positive LD, which shows that variants are less likely to occur on the same haplotype than expected under independence. Importantly, population genetic models without drift do not predict an excess of negative LD. Specifically, by considering Fisher's fundamental theorem, as long as selection is able to efficiently remove load and Muller's ratchet is operating slowly compared to the timescale of coalescence, excess negative LD is not expected as a result of linked selection [23]. For instance, a population with a single non-recombining chromosome with deleterious mutations that are removed at the same rate at which they are created (in mutation-selection balance), does not experience interference [23,28].

Hill and Robertson first studied the effect of linkage disequilibrium on natural selection with only two additive loci under natural selection [21]. They reported that interference creates a detectable excess of negative LD with biologically relevant effective population sizes and variants with realistic selection coefficients. However, their simulations and theory did not incorporate new mutations or multiple loci that differed in age and distance from focal mutations. Adding to their work, McVean and Charlesworth [25] studied the effects of Hill-Robertson interference between weakly selected mutations. They simulated multiple weakly selected mutations and observed that weak selection Hill-Robertson interference generates negative LD between beneficial mutations. This excess of negative LD was most apparent among alleles that were physically close to each other and appeared to disappear with increasing distance between markers. They suggested that interference is a prevalent force driving the distribution of biased codon usage in *Drosophila*. Although they looked at the effects of interference caused by negative selection on fixation probabilities, heterozygosity, and average time to loss, they did not examine the impact of interference among deleterious sites on LD. Additionally, Comeron and Kreitman showed via simulations that interference among multiple positively selected variants should create an excess of negative LD among low frequency variants while reducing overall levels of neutral polymorphism [26]. However, these simulation studies considered only populations of constant size, and their applicability to the LD patterns of natural populations with more complex demographic histories and multiple sites under negative selection remains understudied. Indeed, recent work has found that the expected pattern of background selection is heavily affected by the demographic history of the population [29,30].

Preceding this work, Comeron *et al.* hypothesized that interference could influence the spatial distribution of putatively beneficial codons [31]. Using forward simulations and biologically relevant recombination rates, Comeron *et al.* proposed that a multi-site model of a finite population, with mutations, selection, and linkage could predict the observed relationship between the magnitude of codon usage bias and coding sequence length observed in natural *Drosophila* populations. Recent literature on Hill-Robertson interference proposes

mechanistically why and when Hill-Robertson interference causes negative LD, both in cases of positive selection in asexually evolving populations [24] and negative selection in asexually evolving and recombining populations [23]. In these works, the authors study how the variance in fitness within "effectively asexual linkage blocks" is important when interference is prevalent.

The role of interference has also been assessed in primates (human, chimpanzee, and rhesus macaque) by quantifying the relationship between recombination and $d_N/d_S$ [32]. Bullaughey et al. [32] suggested that there is no detectable effect of recombination on rates of protein evolution. Later, Hussin et al. [33] looked for signatures of interference by quantifying the relative enrichment of deleterious mutations in cold spots of recombination. Theory suggests that cold spots of recombination should be enriched with slightly deleterious mutations relative to hotspots of recombination [34,35]. They identified this relative enrichment in exons, though the strength varied across populations. Additionally, they observed that conserved exons in recombination cold spots are enriched with haplotypes with two NS variants relative to exonic haplotypes in hotspots of recombination. Hussin et al. concluded this excess of mutational burden in cold spots would be expected under a process similar to Muller's ratchet [34]. Here, the least-loaded haplotype class cannot be quickly regenerated if recombination is scarce because it is degraded by drift or new mutations. These results suggested that interference might play a significant role in determining patterns of genetic diversity in human autosomes. Although this study examined the effects of negative selection on the distribution of burden and enrichment of deleterious variation across haplotypes and recombination rates in the human genome, it did not explicitly study the effect of negative selection on LD summary statistics ($r^2$, D, D').

In the present study, we first examined how negative selection affects levels of LD among deleterious nonsynonymous (NS) SNPs relative to the LD among neutral synonymous (S) SNPs using forward simulations. Although all summary statistics of LD depend on allele frequency, we controlled for this possible confounder in our comparisons by limiting pairwise LD calculations using the method of frequency matching described by Eberle et al. [36]. We found in forward simulations with a human-oriented recombination rate, mutation rate, and distribution of fitness effects of new mutations, that negative selection induces a detectable excess of negative LD among derived alleles. Next, we used human data from the Phase 3 1000 Genomes Project (1KGP) [37] and tested for a difference in LD patterns between NS and S SNPs. We found that pairs of derived NS variants tend to be located on different haplotypes (i.e. have more negative LD between them) compared to matched pairs of derived S variants. Additionally, to replicate our results, and to provide a method to detect interference in unphased data sets, we introduce a new summary statistic $H_R^{(j)}$. Using this statistic, we demonstrate signatures of interference in the New York Genome Center's (NYGC) unphased high coverage resequencing of 1KGP individuals [38]. Our findings of an excess of negative LD between pairs of putatively deleterious variants suggest that interference might play a pervasive role in shaping patterns of LD between proximal NS variants in the human genome.

## Results

### Forward simulations

We performed forward simulations using SLiM 3.0 [39]. Each generated chromosome was approximately 5 Mb long and contained intergenic, intronic, and exonic regions. Only NS mutations within exonic regions experienced negative selection (see Materials and Methods). In our simulations, for every 1 synonymous (S) mutation, there are 2.31 NS mutations. This value is derived from degeneracy of the codon table of eukaryotes [40]. We refer to mutations

as NS and S throughout this study because these terms refer to the precise types of mutation analyzed in empirical data. We simulated under three demographic models (**S1 Fig**). Model 1 consists of a population of 10,000 individuals evolving for 100,000 generations. Model 2 is the model of human demography by Gravel *et al.* [41] and implemented into SLiM [39]. Model 3 is identical to Model 2 except that there is no migration across the populations. For each simulation replicate, we sampled 50 individuals from the African population and computed the various LD statistics as described in Materials and Methods.

## Predicted effect of negative selection on LD between pairs doubletons

We first used our simulations to investigate the effect of negative selection on patterns of LD. It is well-known that LD summary statistics are influenced by allele frequency [3,36]. Therefore, we controlled for this effect in the simulations by only considering pairs of SNPs with the same allele frequency in the sample. Further, we only calculated LD statistics between pairs of SNPs with the same functional annotation. For example, we computed LD only between pairs of NS SNPs or between pairs of S SNPs. See **S1 Text** and **S2 Fig** for further simulation results considering other LD statistics and other variant frequencies.

We begin by focusing on doubletons (derived variants that show up in our sample of 50 diploid individuals twice, or variants with a frequency in our sample of 2/100) because we hypothesized that doubletons would be enriched with polymorphisms that are more strongly influenced by negative selection relative to higher frequency variants [42–44]. Consistent with this hypothesis, simulations under Model 1 with a gamma distributed DFE (see Materials and Methods) show that deleterious NS variants with a frequency greater than 2/100 in a sample of 50 individuals tend to be less deleterious than doubletons (**S3 Fig**). Additionally, relative to singletons, we expect doubletons to be less influenced by sequencing error or errors in variant calling in empirical data. From a theoretical perspective, studying doubletons is advantageous because they are the most prevalent type of variants after singletons under the standard neutral model [45]. We quantify LD between derived doubletons using $D'$ rather than $r^2$ because the sign of $D'$ is informative regarding whether the derived alleles preferentially occur on the same haplotypes (are in coupling, also called positive LD) or different haplotypes (are in repulsion, also called negative LD) (see **S4 Fig**). If a pair of doubletons only occur on the same haplotype in our sample of 100 chromosomes, they will have a $D'$ value of 1. Pairs of derived doubletons that never occur on the same haplotype in our sample have a $D'$ value of -1. Thus, the average value of $D'$ at a given distance reflects the number of pairs of SNPs that occur on the same haplotype (more positive) or on different haplotypes (more negative).

We found that $D'$ between doubletons located within 10kb of each other is greatly affected by negative selection (**Fig 1**). When assuming additive effects on fitness, $D'$ for weakly and moderately deleterious ($s = -0.0001$, $s = -0.001$, $s = -0.01$) variants is lower than those for neutral doubletons, indicating that deleterious doubletons ($s<0$) tend to occur on different haplotypes more often than SNPs that are neutral in these simulations (dark blue curve in **Fig 1**). Similar trends are seen when computing other summary statistics of LD (e.g. $r^2$, see **S5 Fig**).

For recessive variants, pairs of moderately deleterious ($s = -0.01$) doubletons also tend to have lower values of $D'$ than pairs of neutral doubletons, when recombination rates are low ($r<1 \times 10^{-8}$ per bp). However, weakly deleterious SNPs ($s = -0.0001$ and $s = -0.001$) have mean values of $D'$ that are greater than those values for neutral SNPs (**Fig 1**). These patterns did not appear to hold across all recombination rates like in the additive case, though the magnitude of the difference is greatest in simulations with low recombination rates. The behavior of recessive deleterious mutations is due to the complex interplay of HRI effects combined with a heterosis effect (associative overdominance) whereby recessive deleterious mutations are masked

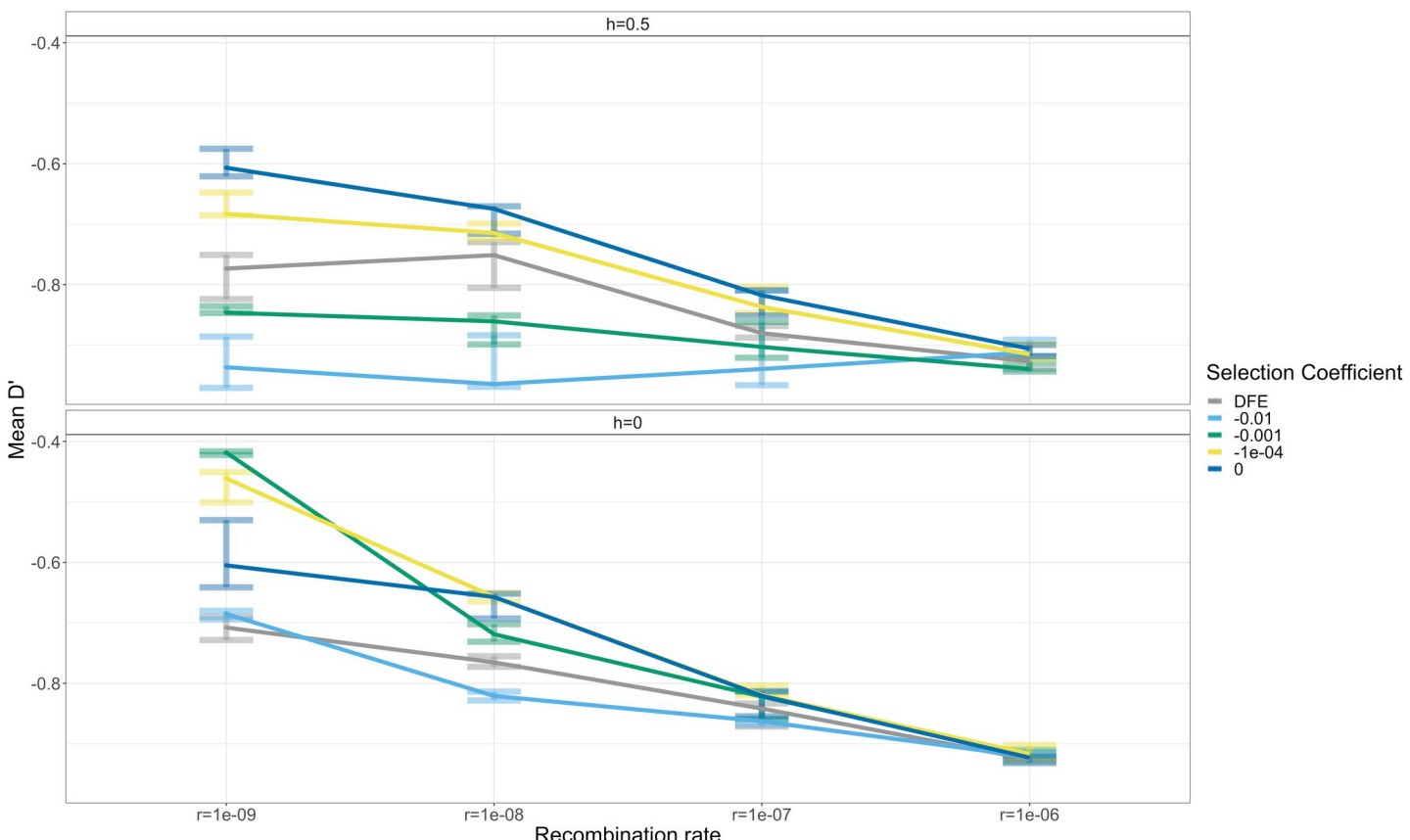

**Fig 1. Mean *D'* for simulated NS doubletons within 10kb from each other across different recombination and dominance parameters.** The differences in the decay curves are most apparent for recombination rate $r = 1 \times 10^{-9}$ per bp and depend on the dominance coefficient of mutations (*h*) and the selection coefficient (*s*) of NS mutations. 150 simulation replicates were simulated for each scenario and were split into 3 groups consisting of 50 simulations. For each group, the mean value of *D'* is shown as the line and the error bars denote the minimum and maximum mean *D'* values.

from selection in the heterozygous state [46]. This effect may be exaggerated if there is little recombination and the deleterious alleles arise on the same haplotype, leading to positive LD.

In real genomes, putatively neutral S and intronic variants are interspersed with deleterious NS mutations. We next compared LD patterns between pairs of NS SNPs (with *s* = -0.0001) to the LD patterns of pairs of neutral S SNPs (no effect on fitness) in our simulations. Consistent with our previous simulations, pairs of derived NS doubleton SNPs (blue line) have less positive LD than pairs of derived S doubletons (yellow line) (**S6 Fig**). This finding suggests that NS doubletons are located on different haplotypes more often than are S doubletons.

As the degree of LD between a pair of doubletons also is influenced by the amount of recombination that occurs between them, we wanted to ensure that the results shown above were not driven by different amounts of recombination occurring between pairs of NS SNPs and pairs of S SNPs. To test for this, we simulated 50 replicates under neutrality but with the same distribution of exons and introns described earlier. We simulated NS SNPs without negative selection by simulating mutations with a NS mutation rate, but with *s* = 0. In summary, the LD patterns for the pairs of NS SNPs without negative selection (red line) matches the LD patterns of pairs of S SNPs (yellow and purple line) (**S6 Fig**), suggesting that the distribution of functional elements is not responsible for the differences in LD patterns between deleterious NS SNPs and interspersed neutral SNPs in simulations.

We also simulated under a more complex demography, Model 2, and found similar trends as the constant size population (see **S2 Text** and **S7–S9 Figs**). Both the simulations with constant selection coefficients for all NS mutations as well as simulations with selection coefficients from a gamma distributed DFE predict deleterious doubletons should have lower values of mean *D'* relative to neutral doubletons.

To examine how LD among variants at other allele frequency categories may be impacted by negative selection, we simulated 3,000 replicates of Model 1 with a recombination rate of 1 x $10^{-8}$ crossovers per base pair per generation, and then computed frequency matched LD statistics (**S1 Table**). Under the assumed simulation parameters, our simulations predict that after deleterious variants increase above a frequency of 6%, they will no longer have lower *D'* values than neutral alleles.

## A new statistic to quantify the effect of selection on the distribution of unphased 2-locus genotypes ($H_R^{(j)}$)

The standard LD statistics $r^2$, *D*, and *D'* are based on the frequency of haplotypes that contain both derived alleles at both loci ($p_{AB}$), which require phased haplotype information. Often when studying nonmodel organisms or unprocessed genotype data, we typically do not have this information and would have to rely on computational phasing. The performance of computational methods to impute or phase variants becomes more challenging for rare variants [47,48], which are the focus of our study. Further, when studying sites influenced directly by natural selection, the assumption of HWE may be violated and the correlation between the diploid allele counts (as denoted by {0,1, or 2}) may not equal the correlation between the haplotypes for two markers.

Here we introduce new statistics that can be directly computed from diploid genotype data to quantify whether derived alleles are more likely to be coupled (i.e. on the same haplotype) or in repulsion (i.e. on different haplotypes) for different functional annotations. Consider a pair of singleton variants. An individual can be homozygous ancestral (0/0,0/0) at both variants, heterozygous at one SNP (0/1,0/0 or 0/0,0/1), or doubly heterozygous (0/1,0/1). If an individual is heterozygous at only one of the SNPs, then by definition, the two singletons must be on different haplotypes. If an individual is heterozygous for both SNPs, it is possible that both derived alleles are carried on the same haplotype. Alternatively, if an individual is heterozygous for both SNPs, it is possible that both derived alleles are carried on different haplotypes (see **S4 Fig**). Quantifying the distribution of homozygous ancestral, singly heterozygous, and doubly heterozygous genotypes does not require information about haplotype phase and thus can be applied to genotype data. We formalize this idea in a statistic that we call $H_R^{(j)}$ (see Materials and Methods for a more detailed description and how to compute $H_R^{(j)}$; also see **S10 Fig**). Our statistic can be calculated from pairs of derived variants at different frequencies in a sample. Applying this statistic to only doubletons in a sample of 50 diploid individuals, corresponding to variants with frequency of 2/100, we call it $H_R^{(2)}$. $H_R^{(1)}$ is used when we compute the statistic on singletons in the sample. Essentially, $H_R^{(j)}$ counts the average number of individuals heterozygous at both SNPs for pairs of SNPs at a given distance apart from each other. This statistic was created to measure the amount of repulsion in a sample, and that is what the subscript "R" stands for. As it is uncommon for doubletons in a sample of 50 to be found in a homozygous state, they are not counted in $H_R^{(j)}$, with little loss of information. Based on predictions from our simulations using other LD statistics (*D*, *D'*, $r^2$), we hypothesized $H_R^{(j)}$ to be lower for deleterious SNPs than for neutral SNPs. To test this hypothesis and examine the behavior of the $H_R^{(j)}$ statistic, we simulated under Model 2 across two different recombination rates ($r$ = 1 x $10^{-9}$ per bp and $r$ = 1 x $10^{-8}$ per bp). Our simulations show that unphased

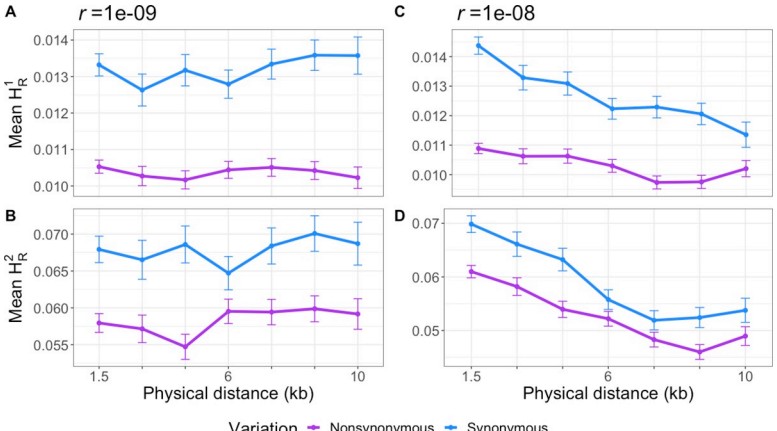

**Fig 2. Distribution of mean $H_R^{(1)}$ and $H_R^{(2)}$ across simulated data with different recombination rates and selection coefficients of NS mutations drawn from a DFE.** Forward simulations predict NS pairs of variants have a mean $H_R^{(j)}$ that is on average lower than that of S pairs of variants. (A) $H_R^{(1)}$ for simulations with a low recombination rate ($r = 1 \times 10^{-9}$ per bp). (B) $H_R^{(2)}$ for simulations with a low recombination rate. (C) $H_R^{(1)}$ for simulations with an average recombination rate ($r = 1 \times 10^{-8}$ per bp). (D) $H_R^{(2)}$ for simulations with an average recombination rate. Each point is the mean $H_R^{(j)}$ statistic in a 1.5kb wide distance bin. Error bars represent one standard error of the mean. 4,300 simulation replicates of Model 2 were simulated for each recombination rate.

genotypes and therefore $H_R^{(2)}$ are affected by negative selection. Specifically, for both recombination rates (**Fig 2**), there is a depletion in mean $H_R^{(2)}$ and $H_R^{(1)}$ for pairs of NS variants compared to S variants. Additionally we computed Spearman's correlation coefficient ($\rho$) between mean $H_R^{(2)}$ and mean $D'$ as well as mean $H_R^{(1)}$ and mean $D'$ (**S11 Fig**). In all cases, the $H_R^{(j)}$ statistics are highly correlated with $D'$ ($\rho > 0.945$, p-value $< 2.2$e-16). Thus, we conclude that $H_R^{(j)}$ can detect interference between deleterious variants using unphased genotypes.

## Matched-pairs permutation test for differences in LD between NS and S SNPs

To quantify differences in LD while controlling for possible covariates, we developed a matched-pairs permutation test on pairs of SNPs (**S12 Fig**). Using the R Package "MatchIt" [49], for each pair of NS SNPs, we extracted a pair of S variants with a similar physical distance between variants (bp), the same allele count between pairs, a similar genetic distance (cM), similar mean *B*-value amongst pairs of variants, and located on the same chromosome as the NS pair. With these matching criteria, we compared pairs of variants that have similar levels of diversity and physical distance between variants. **S13 Fig** shows how closely we required the pair of S SNPs to match the pair of NS SNPs across these characteristics. We then computed the difference between the LD statistic (we considered several different LD statistics, including $D'$ as well as our new $H_R^{(j)}$ statistic) for the NS pair and the S pair. The mean of these differences across all the matched pairs was then computed and used as the test statistic. A matched-pairs permutation test was then conducted with 10,000 permutations. In each test, the null hypothesis was that the mean difference in LD statistics between pairs of NS and S SNPs was 0. The alternative hypothesis was that the mean difference in LD statistic across pairs was less than 0 (the LD statistic among NS pairs was less than that of the matched S pairs). To evaluate the performance of our matched-pairs permutation test, we simulated genomes under Model 1 with simulations that: 1) included negative selection using the distribution of fitness effects inferred by Kim et. al [40], and 2) did not contain negative selection. While a significantly more negative mean $D'$ was seen for NS SNPs in simulations with negative selection

(p < 0.005; matched pairs permutation test), this significant difference was not seen in simulations without negative selection. Additionally, in the simulations without negative selection, we performed this matched-pairs permutation test on other summary statistics of LD and observed no significant difference between pairs of NS variants and S variants (**S14 Fig**).

## Comparison of LD between NS and S SNPs in human data

We next tested whether negative selection has affected patterns of LD across the human genome. Here we first examined 50 phased diploid genomes from the Yoruba (YRI) population of the 1KGP [37]. We computed $D$' for pairs of low-frequency (in this study frequency equal to or less than 5/100, or allele count equal to or less than five) NS variants matched for allele frequency. The same was done with pairs of low frequency S variants. To test for a significant difference in the amount of LD between NS and S SNPs, we used the permutation test described in the previous section where we matched each pair of NS SNPs to a pair of S SNPs that had the same derived allele frequency, physical distance between the pair of SNPs, genetic distance between the pair of SNPs, and magnitude of background selection (See Materials and Methods and **S13 Fig**).

We observed a difference in the mean $D$' of matched pairs of NS and S variants across genetic distance bins (**Fig 3A**). Specifically, NS SNPs have lower values of $D$' at a given distance compared to S SNPs. Application of the matched-pairs permutation test showed that the observed difference of mean $D$' between NS and S variants is significant (p < 0.0005) (**Fig 3B**). These results suggest that pairs of derived alleles at NS SNPs tend to be located on different haplotypes more often than are derived alleles at pairs of S SNPs. Because of our matched-pairs permutation procedure, we conclude the differences in patterns of LD between pairs of

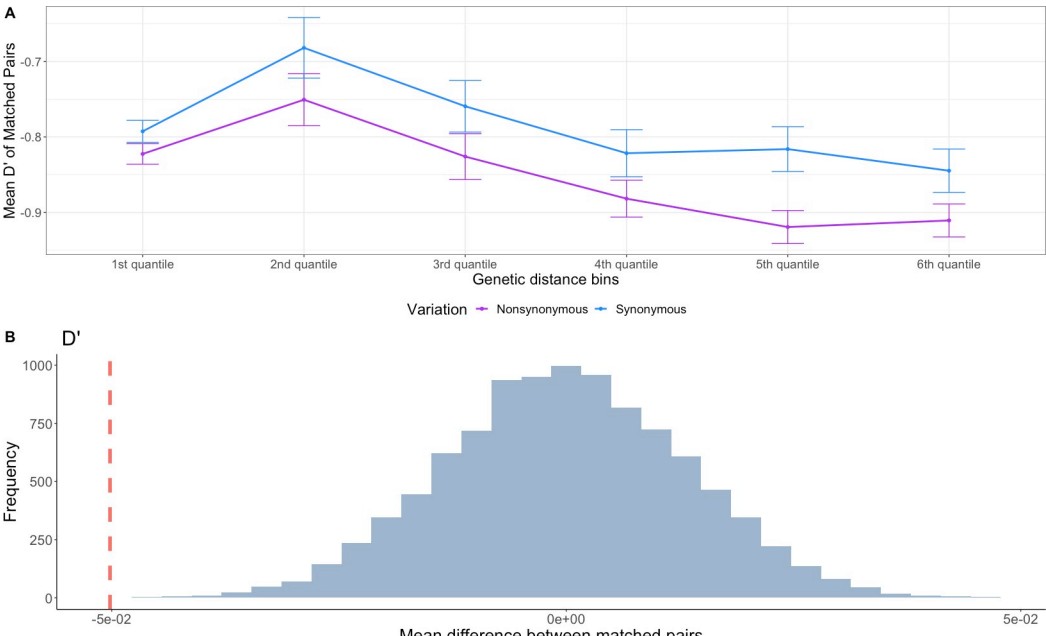

**Fig 3. Pairs of NS variants have lower LD compared to pairs of S variants in human genetic variation data.** (A) Pairs of low-frequency (variants with minor allele count < = 5 in a sample of 50 individuals) NS variants in the YRI population of the 1KGP have a lower (i.e. more negative) mean $D$' compared to pairs of S variants across genetic distance bins. Error bars represent one standard error of the mean and were calculated by dividing the standard deviation by the square root of the number of pairs of SNPs. (B) In the YRI population of the 1KGP, NS pairs of variants have a significantly lower $D$' compared to matched pairs of S variants.

NS SNPs was not due to differences in local recombination rates or differences in background selection.

## Analysis of additional populations

We next tested for differences in LD patterns between pairs of NS and S variants in other human populations. We selected data from 50 individuals from four other non-African populations included in the 1KGP (Han Chinese in Beijing, China (CHB), Utah Residents with Northern and Western European Ancestry (CEU), Mexican Ancestry from Los Angeles USA (MXL), Japanese in Tokyo, Japan (JPT)) and performed identical analyses to those described above to quantify the difference in LD between the different types of SNPs. As in the YRI population, we observed that mean $D$' between matched pairs of NS SNPs was clearly lower than that for pairs of S SNPs across genetic distance bins in three out of four populations (CHB, MXL, JPT; **Fig 4A**) and that these differences were statistically significant in three out of four populations using the matched-pairs permutation test (**Fig 4B**). In the CEU, $D$' was more negative for pairs of NS variants (-0.7806) compared to pairs of S variants (-0.7722), though the difference was not significant. These findings illustrate that across several human populations, NS derived alleles tend to co-occur on different haplotypes more frequently than do S variants that have the same frequency.

## Replication in high-coverage sequence data

Our previous analyses used 1KGP Phase 3 data which is the product of genotype imputation and statistical phasing. To mitigate the possible effects of imputation and haplotype phasing errors on our analysis, we also examined the distribution of NS and S variants using unphased genotypes and our new $H_R^{(2)}$ and $H_R^{(1)}$ statistics to test for differences in LD between NS and S variants. In addition to using the 1KGP Phase 3 dataset, we also used the 30X WGS of the 1000 Genome Project samples, sequenced by the New York Genome Center and funded by NHGRI (http://ftp.1000genomes.ebi.ac.uk/vol1/ftp/data_collections/1000G_2504_high_coverage/working/20190425_NYGC_GATK/) [38].

Our simulations predict that $H_R^{(2)}$ and $H_R^{(1)}$ ought to be lower for pairs of NS variants compared to pairs of S variants (**Fig 2**). In the 1KGP Phase 3 YRI data, on average, pairs of NS variants have a lower mean $H_R^{(2)}$ and $H_R^{(1)}$ (**Fig 5A** and **5B**) compared to S variants. This difference is statistically significant for both doubletons (observed difference = -0.017, permutation test p = 0.011) and singletons (observed difference = -0.006, permutation test p = 0.001). Importantly, the 30X WGS of the same 1000 Genome samples sequenced to higher coverage by the New York Genome Center replicates this result (**Fig 5C** and **5D**). In this data set, the observed difference in mean $H_R^{(2)}$ for NS and S variants is -0.065 (p = 0.0088), and the observed difference in mean $H_R^{(1)}$ is -0.018 (p = 0.001). Thus, our finding that NS SNPs tend to be located on different haplotypes more often than S SNPs is robust to the specific dataset used and is not due to phasing and imputation errors in the original 1KGP Phase 3 data.

## Can Hill-Robertson effects account for more negative LD between NS SNPs?

We hypothesize that Hill-Robertson effects are more likely to impact pairs of variants close to each other, whereas other forces, like synergistic epistasis act on pairs of variants further apart [18]. To better compare the difference between NS and S SNPs as a function of distance, we normalized the difference in $D$ between NS and S pairs of SNPs by dividing by the mean $D$ of S SNPs (**Fig 6**). Overall, this statistic is negative in the empirical data across the entire range of genetic distances considered (green line in **Fig 6**). Simulations under Model 2 show a similar

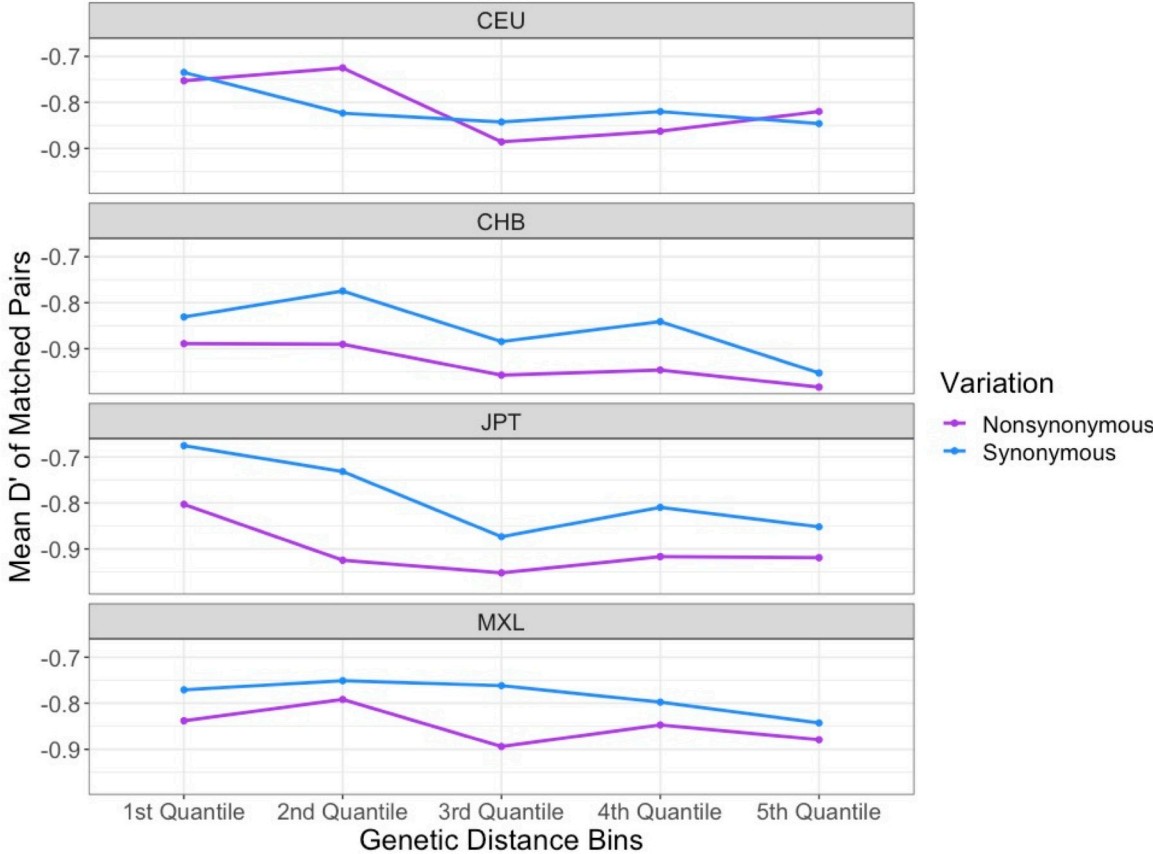

**A**

**B**

| Population | # of matched pairs | NS Mean D' | S Mean D' | Difference (NS - S) | P-Value |
|:---:|:---:|:---:|:---:|:---:|:---:|
| CEU | 1,080 | -0.7806 | -0.7722 | -0.0084 | 0.3421 |
| CHB | 1,405 | -0.9089 | -0.8434 | -0.0655 | 0.0002 |
| JPT | 826 | -0.8484 | -0.7319 | -0.1166 | 0.0001 |
| MXL | 2,963 | -0.8448 | -0.7788 | -0.0660 | 0.0001 |
| YRI | 3,502 | -0.8394 | -0.7892 | -0.0502 | 0.0001 |

**Fig 4. Results from the matched-pairs permutation test for non-African 1KGP populations.** (A) Across four other populations in the 1KGP, NS pairs of variants have lower values of *D'* (i.e. have more negative LD) than their matched S counterparts. The quantile bins on the x-axis were set by combining all pairs of variants across all populations and assigning them into 5 bins based on the centimorgan distance between variants with roughly equal numbers of pairs in each bin. Thus, each population has the same centimorgan distance used to define each bin. (B) Matched-pairs permutation test shows significantly more negative average LD between pairs of NS SNPs than S SNPs in all populations except the CEU population, where this difference is not significant.

pattern, except that the normalized *D* tends toward 0 with increasing genetic distance between SNPs. To assess whether the difference between the simulations and empirical data could be due to the fact that the empirical data has fewer pairs of SNPs (6,066) than do the simulations, we resampled 6,066 pairs from the simulations with the same allele frequencies as those in the empirical data. Here, the normalized difference in LD calculated from the empirical data falls within the range of values seen in the simulations. Some of the non-monotonic trends observed in our analysis of empirical data may also be due to the limited number of SNP pairs in particular recombination rate bins leading to imprecise measurements of the mean *D*. **Fig 6**

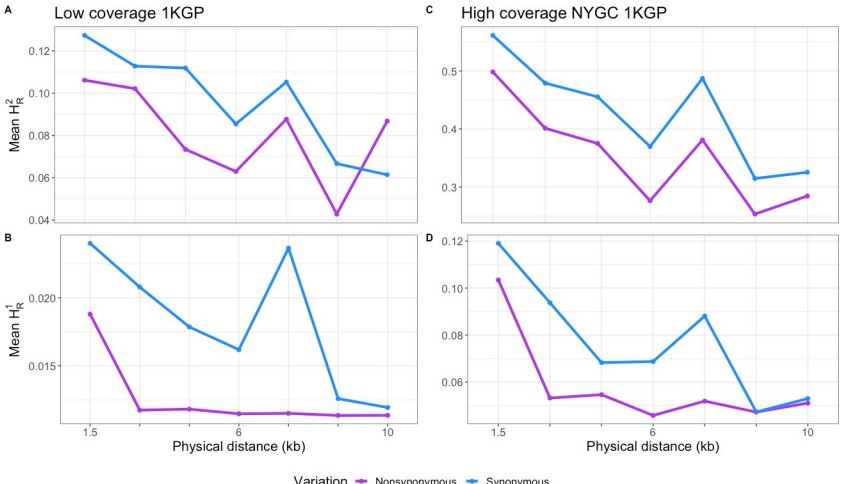

**Fig 5. Distribution of mean $H_R^{(2)}$ and $H_R^{(1)}$ across empirical data sets with different coverages.** (A, B) Low-coverage 1KGP data. (C, D) High-coverage NYGC 1KGP data. For both datasets, mean $H_R^{(2)}$ and mean $H_R^{(1)}$ between NS pairs of doubletons is on average lower than that of S pairs of variants.

shows how downsampling the simulation replicates to have the same number of SNP pairs as do the empirical data (3,160 pairs of NS SNPs and 2,906 pairs of S SNPs) induces fairly high variance, consistent with the empirical observations. Thus, simulations including negative selection can recapitulate the difference in LD between NS and S SNPs at different intervals of genetic distance. We also examined the relationship of mean normalized difference in *D* in CEU and CHB and observed a similar relationship with genetic distance (**S15 Fig**).

Since our forward simulations consisted of mutations that only affect fitness multiplicatively across loci, we conclude that the excess of negative LD between NS SNPs can be explained by interference. However, as a limitation of our study, since we did not explicitly test for it, we cannot rule out a contribution from negative synergistic epistasis in the empirical data (see Discussion).

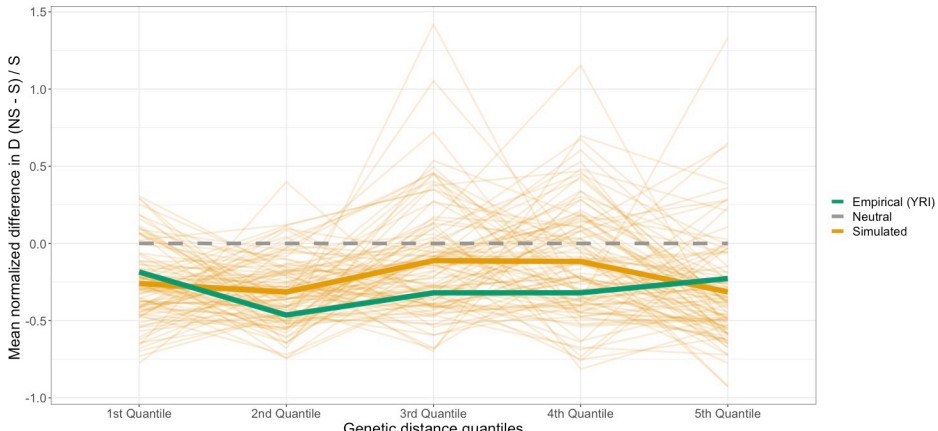

**Fig 6. Mean normalized difference in *D* across genetic distance quantiles.** Empirical (green) and simulated data shows a deficit in *D* between NS variants. The lighter orange lines show 100 resamples of the simulated data. Each resample of the simulated data has the same number of variants with the same allele count as the empirical data.

## Discussion

Here we have shown that patterns of LD are influenced by deleterious mutations, in addition to recombination and the underlying demographic history of a population. In our simulations, we show how Hill-Robertson interference is expected to influence LD patterns between deleterious mutations and find that the effects are likely to be stronger in regions of lower recombination (**Figs 1** and **S2**), consistent with previous work [25,33,50]. Using a human demographic model and DFE, our simulations suggest that interference should be detectable in both low recombination regions ($r = 1 \times 10^{-9}$ per bp) and regions with an average recombination rate ($r = 1 \times 10^{-8}$ per bp) in the human genome (**Figs 1** and **S2**). We then searched for these patterns in several genomic datasets from several human populations and with different types of data (low-coverage and exome sequence data followed by imputation as well as high-coverage whole-genome sequence data). To quantify the difference in LD among NS variants relative to S variants, we implemented a matched-pairs permutation test. Overall, for most of the comparisons considered, NS SNPs appear to be present on different haplotypes more frequently than S SNPs having the same allele frequency (**Figs 3**, **4** and **5**).

Recent studies have identified batch effects present in the 1KGP that have led to population-specific artifacts [51]. We hypothesized that if the excess of negative LD among pairs of NS variants is due to batch effects, we would not detect this difference across multiple populations. However, we see qualitatively similar patterns in 4 of the 5 populations from the 1KGP. Additionally, previous work has suggested that low-frequency errors that are the consequence of batch effects tend to be in positive LD with each other (i.e. the erroneously called derived alleles at different loci tend to co-occur in the same individual) [52]. Also, variants that are identified as error candidates are more likely to be NS variants [52]. With this in mind, we suspect that sequencing errors would bias our analysis of pairs of low frequency variants annotated as being NS towards being more often in positive LD than in negative LD. This suggests that our observation of a depletion of positive LD amongst NS variants is conservative. Lastly, the fact that we replicate these findings using unphased genotypes for $H_R^{(2)}$ and $H_R^{(1)}$ from high coverage sequencing (**Fig 5**) suggests our conclusions are not due to artifacts in the 1KGP.

While we use simulations to quantify the effects of interference on LD statistics, a limitation of this study is that we do not simulate full human genomes and directly assess whether the model parameters can fully explain the empirical data. The DFE that we simulated under closely recapitulates empirically observed site frequency spectrum in human genetic variation data [40]. However, for tractability, we did not incorporate empirically measured recombination maps into our simulations, and instead simulated under uniform recombination rates. Because of this, the LD decay curves in our sample may not quantitatively match the curves seen in empirical data. With these limitations in mind, we compared trends from these simulations qualitatively to our empirical data. Future work can test whether realistic genetic maps, functional annotations, and a DFE can match genome-wide patterns of LD. We focused on variants with minor allele count less than five because in our sample size of 50 they are affected by selection (**S3 Fig**). In many modern datasets, sample sizes are much larger and have many more than 50 individuals. In such datasets singletons and doubletons can be at very low population allele frequencies. In these datasets it may be better to focus on other allele frequency categories for detecting HRI.

Our work adds to the growing literature indicating Hill-Robertson interference is a non-negligible force involved in the spatial distribution of NS variation in the human genome. Sohail *et al.* [18] found that putatively deleterious loss of function mutations were under-dispersed compared to putatively neutral S variation. Their summary statistic essentially

quantified an increase in negative LD between loss of function variants. Their work is similar to our finding of an excess of negative LD among NS variants compared to S variants. However, because the variants they examined were predominantly on different chromosomes and not physically linked, Sohail et al. mainly attributed this excess of negative LD as a signature of synergistic epistasis among deleterious variants. Hussin et al. [33] looked for signatures of interference by quantifying the relative enrichment of deleterious mutations in cold spots of recombination. Our simulations are concordant with this finding and predict that differences in LD among deleterious variants will be most distinguishable in regions of low recombination ($r = 1 \times 10^{-9}$ per bp) and almost indistinguishable in regions of high recombination ($r = 1 \times 10^{-6}$ per bp) (**Fig 1**).

Our results suggest that inferences of demography that depend on genome-wide LD patterns may be biased if negative selection is not accurately modeled. For example, Tenesa et al. used $r^2$ at different genetic distances to infer changes in population size and fit exponential functions to LD decay to infer admixture times [6]. If variants under negative selection are included in these analyses, they may bias parameter estimates. Further, demographic models fit to the site frequency spectrum in humans and *Drosophila* do not recapitulate empirically observed LD patterns [53–55]. Unmodeled negative selection may contribute to this lack of fit. We recommend carefully filtering regions of the genome that may be affected indirectly (linked selection, interference, background selection) or directly by selection by removing loci that have a short recombination distance to functionally annotated loci [56,57]. Alternatively, when considering variants under selection, we find that different combinations of recombination rate, DFE, and dominance predict unique mean $r^2$ and $D$ decay patterns. Thus, LD patterns may offer a strategy to infer the relationship between the dominance coefficient and the DFE in outcrossing populations [58]. In order to accurately predict the expected LD decay curve under a given scenario, extensive forward simulations, or recently developed numerical approaches [59], are needed to determine the predicted LD decay as a function of these genomic and selective parameters.

Further, through methods like LD score regression, LD patterns have been used to learn about the architecture of complex traits [60]. We have shown that for low-frequency variants, variants under selection will have more negative LD than putatively neutral variants (**S1 Table**). This effect may bias methods that assume variants of the same allele frequency will have the same distribution of $r^2$ values. For example, stratified LD score regression is used to infer heritability in different functional annotations and assumes that there will be similar LD between causal variants and tag SNPs regardless of the fitness effects of the causal variants [61]. Because causal variants under greater negative selection may have larger effects on the trait [62], the heritability that deleterious variants account for may be systematically underestimated.

Epistatic interactions have been documented between variants in different genes [63] as well as variants within the same gene [64] and are believed to be widespread properties of biological networks [65,66]. Although the genomic consequences of additive deleterious variants which multiplicatively (across loci) affect fitness are extensively studied, pairwise interactions among variants might be a non-negligible force governing molecular evolution. Current methods to identify plausible pairwise interactions between SNPs rely on classic population genetic summary statistics of LD or other summaries of pairwise association frequencies [65,67,68]. Although we detect differences in LD summary statistics in the human genome among variants of different annotations, our simulations suggest this difference can be explained by interference without the need to invoke epistasis (**Fig 6**). We hypothesize that the difference is predominantly due to interference because of its relationship with recombination rate. Our simulations and theory predict that the biggest differences in LD should be present among

variants with the least amount of recombination separating them, and the smallest differences in LD should be present between variants with the most amount of recombination separating them. This scenario aligns with what we observe in our empirical data. We propose that when developing methods for detecting synergistic epistasis, null models should incorporate Hill-Robertson interference.

Future work could quantify the prevalence of epistasis among linked deleterious variants while jointly quantifying the fitness effects of pairs of deleterious variants [59]. Additionally, although it has been shown that two locus statistics such as *D'* can be used in the detection of epistatic interactions and interference [17,50], it is possible that a combination of summary statistics can better discern between the two. Machine learning approaches, which can combine different features of genetic variation data [69], may provide a powerful tool for detecting salient spatial features of proximal variants involved in epistatic interactions and interference.

## Materials and methods

### Forward simulations

We performed forward simulations using SLiM 3.0 [39]. Each generated chromosome was approximately 5 Mb long and contained intergenic, intronic, and exonic regions. Only NS mutations within exonic regions experienced negative selection. The distribution of genomic elements in our forward simulations followed the specification in the SLiM 4.2.2 manual (7.3), which is modeled after the distribution of intron and exon lengths in Deutsch and Long [70]. Within exonic regions, NS and S mutations were set to occur at a ratio of 2.31:1 [71], which is derived from degeneracy of the codon table of eukaryotes. While intergenic deleterious variation exists, our simulations only considered "exonic" variation. For simulations using a DFE, the selection coefficients (*s*) of NS mutations were drawn from a gamma-distributed DFE with shape parameter 0.186 and expected selection coefficient E[*s*] = -0.01314833 [40]. All NS mutations were either additive with *h* = 0.5 or recessive with *h* = 0.0. The per base pair per generation recombination rate was constant across each simulated region and was fixed at $r \in \{10^{-6}, 10^{-7}, 10^{-8}, 10^{-9}\}$ while the per base pair per generation mutation rate was set to $\mu = 1.5 \times 10^{-8}$.

We simulated under three demographic models. Model 1 consists of a population of 10,000 individuals evolving for 100,000 generations. Model 2 is the model of human demography by Gravel *et al*. [41] and implemented into SLiM [39]. Model 3 is identical to Model 2 except that there is no migration across the populations (**S1 Fig**). We simulated the total number of individuals in the population without additional scaling of parameters with the exception of **S14 Fig**, where population sizes were scaled to be 10-fold smaller. For each simulation replicate, we sampled 50 individuals from the African population and computed the various LD statistics as described in Materials and Methods. Simulation code is on Github for each model.

### Empirical data

We first used data from 50 Yoruba (YRI) individuals from the 1KGP. Specifically, we randomly sampled 50 individuals from Supplementary Table 4 in Gazal *et al*. [72] that were labeled as coming from the "YRI" population, had a mating type described as "OUT" (outcrossing), and had a Q-score (quality score) of greater than 50. Then we removed all non-biallelic variants. Next, we polarized the remaining variants using only high-confidence sites in the 6-way primate EPO multiple alignment as the ancestral allele. All variants without a high-confidence ancestral allele were also removed from analysis. Remaining biallelic exonic variants in our sample were annotated as either NS or S variants using ANNOVAR [73]. For analyses with other populations of the 1KGP Phase 3 data, the same procedure described above was used.

Analysis of the YRI NYGC 1KGP whole-genome sequence data used the same 50 individuals that were selected from the 1KGP Phase 3 data. Variants were also polarized using only high-confidence sites in the EPO multiple alignment as the ancestral allele. All variants without a high-confidence ancestral allele were also removed from analysis. Remaining biallelic exonic variants in our sample were annotated as either NS or S variants using ANNOVAR [73]. Pairs of variants that were within 10,000 bp of each other, had the same allele count in the sample of 50, had the same annotation, and were both located in the 1KGP strict mask (http://ftp.1000genomes.ebi.ac.uk/vol1/ftp/data_collections/1000_genomes_project/ working/20160622_genome_mask_GRCh38/StrictMask/20160622.allChr.mask.bed) were then analyzed. The total number of pairs of variants included in each empirical analysis is indicated in **S2 Table**.

## Computing LD summary statistics with frequency matching

Three different pairwise LD statistics were calculated between SNPs with the same allele count in our sample. First, to compute $D$, we used the formula from Lewontin [17]:

$$D = p_{AB} - p_A p_B.$$

Here, $D$ is the difference between the observed frequency of haplotype $AB$ ($p_{AB}$) and the expected frequency of haplotype $AB$ (assuming random association of the alleles at the two loci, $p_A{}^*p_B$). In this calculation, $p_A$ and $p_B$ are the observed frequencies of the derived alleles $A$ and $B$ in our sample. With the notation described here, the coupling haplotypes include derived alleles at both variants ($AB$) and the repulsion haplotypes include derived alleles at one variant ($Ab$ are $aB$).

We computed $r^2$ with the formula:

$$r^2 = \frac{D^2}{p_A p_a p_B p_b}.$$

$D$' was computed with:

$$D' = \begin{cases} \dfrac{D}{\min(p_A p_B, p_a p_b)} & D < 0 \\ \dfrac{D}{\min(p_A p_b, p_a p_B)} & D > 0 \end{cases}.$$

Whether $D$' is negative or positive depends on the arbitrary choice of the alleles paired at two loci. We chose the pair of derived alleles to be the pair of alleles that cause $D$' > 0 when located in coupling. Therefore, a pair of derived doubletons that only ever appear in a sample together on the same haplotype will have a $D$' = 1. A pair of derived doubletons that are never observed on a haplotype together in a sample will have a $D$' = -1.

Limiting LD calculations between SNPs to restricted allele frequency intervals was first done by Eberle et al. [36] and found to be a more sensitive measure for assessing the average decay of LD and is able to generate average $r^2$ values across nearly the entire informative range. We applied this approach to pairs of NS SNPs as well as pairs of S SNPs. For plotting smooth LD decay curves (**Figs S2**, **S6**, and **S7**), we used a generalized additive model with the "gam" function from the R package *mgcv* version 1.8–29 [74]. We fit the default formula y ~ s (x, bs = "cs"). This formula specifies a generalized additive model between LD ($y$) and physical distance ($x$) with a penalized cubic regression spline which has had its penalty modified to shrink towards zero (bs = "cs") [74].

## A new LD statistic for unphased genotypes: $H_R^{(j)}$

We developed a new statistic to be applied to unphased genotype data to quantify whether derived deleterious alleles at a pair of SNPs are likely to be found on the same haplotype. We call our new static $H_R^{(j)}$, where the "R" stands for "repulsion" as we use this summary statistic to measure repulsion between deleterious variants.

We begin by computing the distribution of counts of homozygous ancestral, singly heterozygous, and doubly heterozygous genotypes amongst pairs of variants that were within 10,000 bp from each other, had an identical allele count, and also had the same functional annotation (S or NS). Here variants can have three genotypes 0 (homozygous ancestral), 1 (heterozygous), 2 (homozygous derived). For example, for a pair of NS singletons in a sample of 50 diploid individuals, the counts could be: $n_{11} = 1$, $n_{01}$ and $n_{10} = 0$, and $n_{00} = 49$. This would correspond to 1 individual being doubly heterozygous for this pair of variants, 0 individuals being singly heterozygous, and 49 individuals being homozygous for the ancestral genotypes at these loci.

Then,

$$H_R^{(j)} = \frac{1}{l_j} \sum_{i=1}^{i=l_j} n_{11}^{(i)}.$$

$H_R^{(j)}$ depends on three components: $n^{(i)}_{11}$, $j$, and $l_j$. $j$ refers to the allele count of variants to be analyzed. For singletons, $j = 1$, for doubletons $j = 2$, for tripletons $j$ would be equal to 3, etc. $l_j$ is the number of pairs of variants at allele count $j$ within a distance threshold. $n^{(i)}_{11}$ represents the count of heterozygous individuals with derived variants at both loci of the pairwise comparison $i$ (coding genotypes as 0 for homozygous ancestral, 1 for heterozygous, 2 for homozygous derived). For pairs of doubletons, this equation is represented as:

$$H_R^{(2)} = \frac{1}{l_2} \sum_{i=1}^{i=l_2} n_{11}^{(i)}$$

and would be called $H_R^{(2)}$. To compute $H_R^{(2)}$ for derived NS variants in a sample, we created a list of all derived NS doubletons in the sample. Then, we created a list of all unique pairwise combinations of these doubletons that involve SNPs within a certain distance threshold. The length of this list would be equal to $l_2$. We then found the number of heterozygous individuals with derived variants at both loci of the pairwise comparison $i$ are then summed together over all $l_2$ pairs of SNPs. This is represented by $\sum_{i=1}^{i=l_2} n_{11}^{(i)}$. Lastly, division by the total number of pairwise comparisons (or multiplication by $\frac{1}{l_2}$) gives $H_R^{(2)}$.

In principle, $H_R^{(j)}$ can be computed for any value of $j < 2n$, where $n$ is the number of individuals in the sample. In practice, we consider $j = 1$ and $j = 2$ because low-frequency variants in a sample size of 50 are most likely to be impacted by negative selection (S3 Fig).

## Estimating genetic distance between variants

The genetic distance between two markers was computed using the high-resolution pedigree-based genetic map assembled by deCODE [75]. First, we averaged the male and female genetic maps. Occasionally the genetic distance between two markers in our sample was not explicitly estimated by deCODE. If one or both of the markers were in regions not measured by deCODE, we removed these markers from our analysis with genetic distance. However, if the two markers were within a region of the genome with a high-resolution recombination environment estimated by deCODE, we imputed the genetic distance between markers by

estimating a centimorgan per base pair rate and multiplying this rate by the physical distance between the two markers that lacked a genetic distance annotation.

## Annotating variants with amount of background selection

For one of our matched-pairs permutation tests, we matched pairs of NS variants with pairs of S variants with similar levels of background selection. *B*-values were downloaded from http://www.phrap.org/software_dir/mcvicker_dir/bkgd.tar.gz. Then we used liftOver (UCSC Genome Browser) with its default settings and the hg18Tohg19 chain (http://hgdownload.cse.ucsc.edu/goldenPath/hg18/liftOver/hg18ToHg19.over.chain.gz) to convert *B*-value coordinates from hg18 to hg19. Each SNP in our data set was then annotated with its corresponding *B*-value using the GenomicRanges package [76].

## Supporting information

**S1 Fig. Demographic models for simulations.** Model 1 represents a constant population size simulation. Model 2 represents the Gravel et. al (2011) demographic model. The yellow sub-population represents Africans (YRI). Green represents the ancestral Eurasian bottleneck. Blue represents East Asia and pink represents Europe. Model 3 is the Gravel demographic model but without migration.
(TIFF)

**S2 Fig. Decay of mean $r^2$ for different recombination and dominance parameters.** The differences in the LD decay curves are most apparent with recombination rate $r = 1 \times 10^{-9}$ per bp and depend on the dominance coefficient ($h$) of mutations and the selection coefficient ($s$) of NS mutations. DFE curves come from simulated populations where new NS mutations have selection coefficients following our defined distribution of fitness effects (see text). All variants are included in this analysis and no frequency filters are applied.
(TIFF)

**S3 Fig. Doubletons in a sample of 50 individuals are more deleterious than higher frequency variants.** We simulated 300 replicates of a constant population size of 14, 474 diploid individuals with $r = 1 \times 10^{-8}$ per bp, and the DFE and genome structure defined in Materials and Methods. Although simulations predict singletons (first red point), on average, should be the most deleterious in our samples, doubletons (second red point) are also relatively deleterious. Higher frequency variants tend to have mean selection coefficients that are much more neutral. Because we want to study the effects of negative selection on LD, we restrict many of our analyses to low frequency variants (allele count $< = 5$)
(TIFF)

**S4 Fig. Pictorial representation of variants in coupling and repulsion.** Pairs of derived variants that co-occur on the same haplotypes more frequently than expected are said to be "coupling" or in positive LD (left panel). Pairs of derived variants that occur less frequently than expected on the same haplotype are said to be in "repulsion" or in negative LD (right panel).
(TIFF)

**S5 Fig. Mean $r^2$ for simulated nonsynonymous doubletons within 10kb of each other across different recombination and dominance parameters.** The differences in the decay curves are most apparent for recombination rate $r = 1 \times 10^{-9}$ per bp and depend on the dominance coefficient ($h$) and the selection coefficient ($s$) of NS mutations. 150 simulation replicates were simulated for each scenario and were split into 3 groups consisting of 50 simulations. For each group the mean value of $r^2$ is shown as the line and the error bars denote the minimum and

maximum mean $r^2$ values.
(TIFF)

**S6 Fig. Decay of $D$ from simulated NS and S doubletons.** There are two different scenarios shown in the figure: 1) simulations with negative selection, and 2) simulations without negative selection (completely neutral evolution). The blue line is the decay curve of NS doubletons with direct negative selection acting on them. In contrast to the decay curve of the NS doubletons not under negative selection (dark red), the NS doubletons with direct negative selection acting on them (blue) have a more negative $D$ than the other neutral doubletons (S variants in simulations without selection, NS variants in simulations without selection, S variants in simulations with selection).
(TIFF)

**S7 Fig. Decay of $D$ for simulated NS and S doubletons in scenarios with migration.** For $r = 1 \times 10^{-8}$ per bp, the largest differences between S and NS $D$ is predicted by our simulations to be within the 0–10,000 bp range. In our simulations with $r = 1 \times 10^{-9}$ per bp, noticeable differences between S and NS $D$ is predicted by our simulations to be across the 0–100,000 bp range. Additionally, migration appears to qualitatively obscure the difference in $D$ between types of variants at intermediate distances.
(TIFF)

**S8 Fig. Mean selection coefficient of deleterious doubletons in our simulated samples and their population of origin.** For both $r = 1 \times 10^{-9}$ per bp and $r = 1 \times 10^{-8}$ per bp, doubletons that appear in our simulated "African" sample, have on average different mean selection coefficients depending on their population of origin. Migration (Model 2) allows for deleterious variants that originated from other populations to appear in our African samples. On average, the doubletons in our sample of African individuals that originate from East Asia and Europe are less deleterious than the doubletons in our sample that originated from Africa.
(TIFF)

**S9 Fig. Simulations under demographic Model 2 predict NS pairs of doubletons are more often in complete repulsion than S pairs of doubletons.** The red line denotes the total fraction of simulated NS doubletons that are in complete repulsion ($D'$ = -1) and the blue line denotes the total fraction of simulated S doubletons that are in complete repulsion. For each recombination rate, pairwise LD computations were binned into 5 quantiles based on the centimorgan distance between variants. The midpoint between the boundaries of each bin was then computed and defined the bins on the x-axis.
(TIFF)

**S10 Fig. Pictorial representation of $n_{AB}$ and $H_R^{(1)}$.** The statistic $H_R^{(1)}$ depends on the number of unique pairwise comparisons among singletons within the distance threshold 10 kb ($l_1 = 1$). Additionally, it depends on the number of individuals who are heterozygous at both loci. In both Scenario 2 and 3, the individual is heterozygous at both loci, thus making $H_R^{(1)} = 1$. The statistic $n_{AB}$ depends on the number of haplotypes that contain derived alleles at both loci. In this case, only Scenario 2 contains one haplotype that has both derived variants. $H_R^{(1)}$ can also be thought of as an indicator variable taking the value of 1 if an individual is heterozygous at both loci and 0 if not.
(TIFF)

**S11 Fig. Spearman's rank correlation coefficient ($\rho$) between $H_R^{(j)}$ and $D'$.** (A) For simulations with $r = 1 \times 10^{-9}$ per bp, $\rho = 0.978$ (p-value < 2.2e-16) for the correlation between $H_R^{(1)}$ and $D'$. (B) Also, $\rho = 0.978$ (p-value < 2.2e-16) for the correlation between $H_R^{(2)}$ and $D'$. (C)

For simulations with $r = 1 \times 10^{-8}$ per bp, $\rho = 0.947$ (p-value $< 2.2e\text{-}16$) for the correlation between $H_R^{(1)}$ and $D'$. (D) Also, $\rho = 0.996$ (p-value $< 2.2e\text{-}16$) for the correlation between $H_R^{(2)}$ and $D'$. The DFE of NS mutations was gamma-distributed with shape parameter 0.186 and expected selection coefficient $E[s] = -0.01314833$.
(TIFF)

**S12 Fig. Flow chart of the matched-pairs permutation test.** First, sample 50 individuals from a population. Polarize variants and then annotate variants as either NS or S. Second, compute LD summary statistics among pairs of variants that are within 10,000 bp from each other and have the same allele count (AC) and annotation (NS or S). Third, for each pair of NS variants with a computed LD statistic, find one S pair of variants with the same AC, on the same chromosome, and with a similar ($<50$ bp) distance between variants. Each pair of NS and S pairs constitutes a matched pair. Fourth, compute the mean difference between matched pairs. Fifth, permutate the label (i.e. S or NS) for each pair of SNPs.
(TIFF)

**S13 Fig. Matching rules for features and their distribution across NS variants and S variants in empirical data.** In order to reduce possible confounding effects of different distances, allele counts, levels of background selection, and recombination rate, we implemented a matched pairs scheme. For each pair of NS variants in our data set, we attempted to identify a pair of S variants with similar features. This allowed us to compare pairs of variants with similar distributions and spread of covariates. (A) Physical distance between pairs of S SNPs vs. pairs of NS SNPs after matching. (B) Distribution of allele counts for NS and S SNPs after matching. (C) Distribution of genetic distances between pairs of NS and S SNPs after matching. (D) Distribution of $B$-values surrounding NS and S SNPs after matching. (E) The number of NS and S SNPs per chromosome after matching. (F) The matching rules and summary statistics of matching success. The "Matching" column denotes the rules we used for matching. "SD" stands for standard deviation.
(TIFF)

**S14 Fig. Matched-pairs permutation test on various LD summary statistics in simulations without negative selection.** (A) $D$. (B) $D'$. (C) $r^2$. (D) Count of the derived haplotype (AB). In simulations without negative selection, none of the LD summary statistics we computed showed a significant difference between NS and S matched pairs.
(TIFF)

**S15 Fig. Mean normalized difference in $D$ across genetic distance quantiles for multiple 1KGP populations.** Empirical and simulated data show a deficit in $D$ between NS variants. The lighter orange lines show 100 resamples of the simulated data. Each resample of the simulated data has the same number of variants with the same allele count as the YRI empirical data. All three populations show a more negative $D$ between low frequency NS variants compared to low frequency S variants.
(TIFF)

**S1 Table. Summary statistics of $D'$ across allele counts for neutral (S) and deleterious (NS) variants.** In simulations with a gamma distributed DFE, NS pairs of low-frequency variants are predicted to have a less positive $D'$ compared to S pairs of variants. Generally, as the frequency of variants analyzed increases, $D'$ becomes more positive and the differences between NS and S pairs decreases.
(TIFF)

**S2 Table. Summary of empirical data used in the figures.** The number of pairs of variants in each analysis is in the "Number of pairs" column. Whether or not matching was required between synonymous and nonsynonymous pairs of variants is indicated in "Matching between S and NS pairs" column. The allele count of variants included in the analysis are indicated in the "Allele count of variants" column. The data set from which variants come from and any notes included are indicated in the "Notes" column. If "Matching between S and NS pairs" is "Yes", then the total number of NS and S pairs are reported in the "Number of pairs" column. (TIFF)

**S1 Text. Predicted effect of negative selection on linkage disequilibrium decay.**
(PDF)

**S2 Text. Effect of negative selection and complex demography on LD.**
(PDF)

## Acknowledgments

We thank Nandita Garud and Jazlyn Mooney as well as members of the Lohmueller lab for helpful discussions throughout the project and comments on the manuscript. We would like to thank the New York Genome Center for producing and making the 30x coverage sequencing of the 1000 Genomes samples available pre-publication.

## Author Contributions

**Conceptualization:** Kirk E. Lohmueller.

**Formal analysis:** Jesse A. Garcia.

**Funding acquisition:** Kirk E. Lohmueller.

**Investigation:** Jesse A. Garcia.

**Project administration:** Kirk E. Lohmueller.

**Software:** Jesse A. Garcia.

**Supervision:** Kirk E. Lohmueller.

**Visualization:** Jesse A. Garcia.

**Writing – original draft:** Jesse A. Garcia, Kirk E. Lohmueller.

**Writing – review & editing:** Jesse A. Garcia, Kirk E. Lohmueller.

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
