## [Decision Letter · Decision Letter 0]

5 Jun 2020

Dear Dr Lohmueller,

Thank you very much for submitting your Research Article entitled 'Negative linkage disequilibrium between amino acid changing variants reveals

interference among deleterious mutations in the human genome' to PLOS Genetics. Your manuscript was fully evaluated at the editorial level and by independent peer reviewers. The reviewers expressed enthusiasm for this work, but raised some concerns about the current manuscript. Based on the reviews, we will not be able to accept this version of the manuscript, but we would be willing to review again a much-revised version. 

If you decide to revise the manuscript for further consideration at PLOS Genetics, please aim to resubmit within the next 60 days, unless it will take extra time to address the concerns of the reviewers, in which case we would appreciate an expected resubmission date by email to plosgenetics@plos.org.

[LINK]

We are sorry that we cannot be more positive about your manuscript at this stage. Please do not hesitate to contact us if you have any concerns or questions.

Yours sincerely,

Hua Tang

Section Editor: Natural Variation

PLOS Genetics

Scott Williams

Section Editor: Natural Variation

PLOS Genetics

Reviewer's Responses to Questions

**Comments to the Authors:**

Reviewer #1: In this manuscript, Garcia and Lohmueller investigate Hill-Robertson interference, which occurs when deleterious variants with additive effects tend to occupy disjoint haplotypes. Overall, this is a convincing and very thorough paper. A particular strength is the comprehensive and well-written Introduction section. I also appreciated the extensive analyses of genetic data from different populations and different types of sequencing data. My only major comment is that the organization of the Results section could be improved.

Major presentation comment

The structure of the Results section is difficult to follow. Right now, there are many rounds of simulations, comprising more than half the methods section, but after these many subsections there is a one-sentence statement that none of the LD metrics computed in simulations can actually be computed on real data due to lack of phased data (I disagree with this statement; see minor comment 7 below). Then, two new methods are introduced (H_R(j) and a permutation test), and it’s not clear what the relationship is between them (I assume the permutation test is applied to H_R(j), but this isn’t stated). Then, results on real genotypes are presented, and at the same time, new simulations are performed to show that the new methods work.

I suggest the following organization for the results section: (0) optionally, describe a small set of motivating simulations that illustrate the effect that you wish to quantify; (1) describe new methods and why they are needed, including H_R(j) and the permutation test; (2) perform simulations, and devote at least one simulation subsection to demonstrating that the new method/metric performs as intended; (3) apply new method/metric to real genotypes.

Minor comments

1. Figure 1: Is the dependence of LD on bp distance and recombination rate fully mediated by the map distance? I.e. is LD approximately the same at distance=1kb and rate=10^-7 vs d=10kb and r=10^-8? If not, please explain why. If so, consider changing the x axis to map distance and reducing the number of figure panels from 8 to 2. Also, please set the minimum y-axis value to zero on all panels.

2. “no consistent ordering of how selection coefficients affect genome-wide r2 decay with physical distance”: what is the intuition for why this occurs?

3. “Human-like exomes… human-like levels of negative selection”: when model parameters are chosen to match human-inferred values, do resulting observable quantities actually match what we observe in humans? Is the allele frequency spectrum similar? Is the average LD similar?

4. “As the degree of LD between a pair of doubletons also is influenced by the amount of

recombination that occurs between them, it is possible that there could be different genetic distances (cM) between pairs of NS and S SNPs”: I don’t understand this sentence. Should it instead state that different genetic distances between NS vs S SNP pairs could be an alternate explanation for differences in LD?

5. Effect of background selection on LD between S variants: this is interesting, and I think it merits further investigation, but Figure S5 does not convince me that the effect is real. Is the slight difference that is observed at d<500bp statistically significant? Is it explained by differences in allele frequency spectrum? What is the selection coefficient for NS variants in these simulations, and might the effect be stronger under weak selection than under strong selection?

6. Proportion of S vs NS variants with D=-1: this long paragraph doesn’t cite any main or supplementary display items, and it doesn’t seem especially well motivated – given that E(D) is already shown to be different, is there any scenario where P(D=-1) would not be different? Given that P(D=-1) is not quantified in analyses of real data, I think that this subsection could be deleted.

7. “traditional LD statistics… require phased haplotype information”: It’s not clear to me that this is true. Let r_phased be the correlation between the haploid genotypes for two markers, and r_diploid be the correlation between the diploid allele counts. Under Hardy-Weinberg, r_phased=r_diploid. Are potential HWE deviations the reason for needing something more sophisticated? Is observable deviation from HWE even possible for doubletons? Overall, I feel that H_R(j) could be motivated more clearly. (However, this manuscript does not need a nontrivial new statistic to be a valuable contribution).

8. D’ values of S vs NS variants: are the observed values, including the apparently non-monotonic trend, consistent with simulations? Are the S values consistent with neutrality?

9. Figure 4: I suggest to calculate standard errors and include error bars for panel A; to use a linear x-axis scale in panel A with round numbers as x-tick values; to reduce the number of subpanels in B from 3 to 1, as all three subpanels convey the exact same information; and to move panel C content to simulations section.

10. Figure 4b presents traditional LD metrics, seeming to contradict the statement on p. 18 that they cannot be computed on real data due to lack of phased information. Are these calculated using computational phasing? Are there any limitations to this approach (e.g. that it cannot be applied to variants below some MAF threshold)?

Reviewer #2: In their manuscript "Negative linkage disequilibrium between amino acid changing variants reveals interference among deleterious mutations in the human genome" Garcia and Lohmueller examine the signal of LD between deleterious alleles. The study first analyses the expected patterns of LD using forward in time simulations, before studying these patterns in empirical human population data. The study shows that LD between deleterious loci is lower than neutral equivalents, suggesting interference. In addition, the authors introduce a new statistics (H_R) that allows to detect interference in unphased genomic data.

Overall, this is a very well conceived study. In nicely combines intuition building through simulations with empirical analyses. The manuscript is timely, well-written and interesting. Few modifications and additional information would make the manuscript more accessible to a broader audience.

Major comments:

- I find it quite difficult to wrap my head around negative LD when values in the figures (e.g., Fig. 1 and 2) are positive. I think it would increase the accessibility of the manuscript present values of D' instead of r2.

- It would be good to see results of LD for higher frequency alleles. Do they show the same pattern as doubletons, or does it change? Some deleterious alleles might actually be at higher frequencies due to demography or introgression. Would be nice to see if there is a threshold for the observed pattern.

- How is the LD calculation and pattern influenced by the difference in diversity between NS and S loci? Did you compare only pairs with equal physical distance?

- While I think a uniform s is a good start and helps to build intuition about the effect, it would be great to have one or two additional simulations with (e.g., gamma) distributions that would allow to track which combination of effect sizes create the pattern. Or at least explore this further in the DFE simulations. What combinations of s create negative LD and which ones not. This would be useful for the interpretation of empirical data.

- The use of H comes a bit out for the blue for me and could maybe be better integrated into the flow. It would be particularly useful the show correlations of other LD measures with H. The fact that you have phased data should not harm H calculations and how does this compare to unphased results?

- How is the nomenclature for H_R determined? Not sure I like the use of "H" in this context as it is used already extensively (Not that many other letters are being used uniquely). If you want to stick to "H" maybe it would be good to include LD as index instead of R to make clear that it's a measure of LD.

- I was wondering how many sites are actually used for each analysis in the empirical data. Could you maybe include a supplementary table that indicates the number of sites used, im particular for the H_R analysis.

- Does NS and S come from "synonymous" and "non-synonymous"? Does this not unnecessarily narrow the sites. The simulations are not limited to those and in reality other loci might be as deleterious (e.g., promotors). I think it would be clearer to use deleterious and neutral throughout the manuscript. In the empirical part the use of non-synonymous as proxy for deleterious can be specified.

Further comments:

Introduction:

- It would be useful to introduce the concept of negative LD in the introduction

- "... or negative synergistic epistasis play a pervasive role in shaping patterns of LD between proximal NS variants in the human genome." This seems to imply that you tested for negative synergistic epistasis, but this is not the case. I would suggest to rewrite.

Material and methods:

- Would be good to give more details on the simulations in the M&M section. Even if some parameter values (N and generations) are mentioned in the Results, I think it wouldn't harm to have them all together in the M&M section. This will make the workflow more reproducible.

- is there an implementation code for H_R that other people could use on their data? Would be great if you could implement it to work on VCF files for others to test those patterns. I think this would increase the impact of your study substantially.

Figures:

Fig. 1 why do strongly deleterious alleles show higher LD than neutral alleles? if it was only for the lack of interference as described in the "results" would we then not expect the no difference to neutral. But observed is an increase. Can you give further explanations for this increase?

Reviewer #3: In this manuscript, the authors use simulations to study how patterns of linkage disequilibrium (LD) are affected by negative selection. They show using simulations that deleterious mutations can be brought into negative linkage disequilibrium by the action of linked selection, i.e. even in the absence of epistatic interactions between the mutations. They show that the excess of negative linkage disequilibrium can be maintained in complex demographic models thought to be realistic for humans, and also demonstrate that an excess of negative linkage disequilibrium can be observed in empirical data in the difference in patterns of LD between nonsynonymous and synonymous variants. They highlight that an excess of negative LD due to linked selection can lead to problematic estimates of quantities inferred from genome-wide LD patterns under the assumption that these are not significantly impacted by linked selection. I think this is an important and valuable point, especially given the central role that these methods play in statistical and population genetics.

Overall, I find the manuscript very interesting and the main conclusions convincing. I believe it represents a valuable contribution to the literature, but I have one substantial concern, which is that I don’t believe that the effect that leads to an excess of negative LD has been accurately described and interpreted. Specifically, there is existing theoretical literature on Hill-Robertson interference causing negative LD that explains why and when Hill-Robertson interference causes negative LD, both in cases of positive selection in asexually evolving populations (see Gomez et al. 2019 Genetics) and negative selection in both asexually evolving and recombining populations (see Good et al. 2014 PLoS Genetics). The latter work in particular shows that negative LD may or may not arise in the presence of linked negative selection, depending on how efficient selection is at purging deleterious mutations from the population. It would be important for the text of the manuscript to be revised to accurately reflect this effect (and to acknowledge these earlier authors describing the effect) — I give details in my first major comment.

Major comments:

1) The argument outlined in the paragraph starting on the bottom of page 5 and continuing on page 6 is not entirely accurate. It attributes the entire effect of negative LD to an increase in relative fitness of single mutants due to the presence of multiple mutants. Yet it is not difficult to show that this effect on its own does not lead to an excess of negative LD. Consider for instance a population with a single non-recombining chromosome in mutation-selection balance, in which deleterious mutations are removed at the same rate at which they are created, so that the population is not experiencing Muller’s ratchet. For simplicity, let’s focus on the case that all mutations have the same effect on fitness, -s, though this is not essential to the argument. In this case, at mutation-selection balance, the population assumes a Poisson-shaped fitness distribution with an average uL/s mutations per individual, where u is the per-site mutation rate, and L the number of sites, as described by Haigh (1978) and others. This means that the deleterious mutations are distributed independently, as if they were freely recombining and excess negative LD does not arise in this case. This is in direct contrast to the explanation given by the authors.

By considering Fisher’s fundamental theorem, one can show that as long as selection is able to efficiently remove this load and Muller’s ratchet is operating slowly compared to the timescale of coalescence, excess negative LD is not expected as a result of linked selection. By Fisher’s fundamental theorem, the rate of change in the mean fitness, v is related to the fitness variance sigma^2, mutation rate and selection coefficient s according to: v = sigma^2 - uLs. Here we see that the Poisson case of efficient selection described above corresponds to v = 0 and sigma^2 = u L s, whereas the ratcheting case has v < 0, and so sigma^2 < u L s, and deleterious mutations tend to be in negative LD. This effect persists in the presence of recombination — see Good et al. 2014 PLoS Genetics, and in particular Fig. S1, and the Supplementary Texts.

Note that this rationale also agrees quite well with the simulation results seen here — there is an intermediate strength of selection at which excess negative LD is strongest, which also agrees with expectations of the rate of Muller’s ratchet being maximized at intermediate fitnesses. However, it does suggest that many of the simple arguments offered here do not stand up to scrutiny.

2) The focus on doubletons to quantify the effects of negative selection is not well justified from a theoretical standpoint (though I agree that using doubletons appears generally better than the common practice of using singletons, as the authors explain). Ultimately, the important population-genetic factor is the frequency of an allele in the population, rather than in the sample, and depending on the sample size, doubletons can represent vastly different frequencies. In many modern datasets, singletons and doubletons can be at such low frequencies they are primarily affected by genetic drift for reasonable strengths of selection, which means that the main effect that the researchers aim to study is likely difficult to observe on theoretical grounds, and possibly attributable to other causes. In general, while I do agree with the authors that patterns of LD are strongly influenced by frequency, the choice of target frequency should be guided by theoretical considerations and parameter values, in addition to heuristics for the elimination of technical artifacts in a dataset (which I agree are also very important).

For instance, in some cases examined here — for instance for s = 0.001, N = 10 000, Ns = 10, which means that all variants below a frequency of 1/Ns = 10% are primarily affected by drift. Doubletons in these samples (n= 100) are expected to be present in the population at frequencies 2% or lower, meaning that on their own, they should not be strongly impacted by selection. In fact, for these parameters, we can quite confidently claim that the effects of negative selection should be very weak at the target frequencies if all haplotypes contain no more than two deleterious mutants. Thus, any effect visible at these frequencies must be attributable to statistical features stemming from linkage to (large numbers of) other deleterious mutations, which reinforces the point above that the causes of negative LD are subtler than described here.

To avoid these technical artifacts of sample size, and understand better the frequency-dependence of the effect describes here, would it be feasible to look at these statistics as a function of the entire frequency spectrum and attempt to identify transition points in ways that are self-consistent with the input parameters?

Minor comments:

1) I would argue that the statement that there has been comparatively less empirical work quantifying Hill-Robertson interference as a bit ungenerous — given how broadly the term is applied, one could argue that many papers attempting to quantify the strength of background selection (including some classical ones by Charlesworth et. al, as well as more recent work by e.g. McVicker et al (2009) PLoS Genetics, Elyashiv et al. (2016) PLoS Genetics) are doing precisely that. Perhaps make the statement a bit narrower to clarify?

2) I think it is a great point that unmodeled negative selection may contribute to differences between empirically observed LD patterns and demographic models fit to the site frequency spectrum (page 26, Discussion). However, I disagree with the suggestion that filtering variants that may be affected by selection should be sufficient, unless nearby neutral SNPs are also filtered. As this manuscript (and others) demonstrate, linkage between deleterious variants is critical to producing negative LD, and this form of linkage also affects patterns of variation at neutral sites.

**Have all data underlying the figures and results presented in the manuscript been provided?**

Reviewer #1: Yes

Reviewer #2: No: Simulation code and simulation summary statistics seem not to be available at the moment. As simulation outcome strongly depends on the exact parameters used, code should be provided as link to a repository or as supplementary data.

Reviewer #3: Yes

PLOS authors have the option to publish the peer review history of their article (what does this mean?). If published, this will include your full peer review and any attached files.

Reviewer #1: Yes: Luke J O'Connor

Reviewer #2: No

Reviewer #3: No

---

## [Decision Letter · Decision Letter 1]

10 May 2021

Dear Dr Lohmueller,

Thank you very much for submitting your Research Article entitled 'Negative linkage disequilibrium between amino acid changing variants reveals

interference among deleterious mutations in the human genome' to PLOS Genetics.

The manuscript was fully evaluated at the editorial level and by independent peer reviewers. The reviewers appreciated the attention to an important topic but identified some concerns that we ask you address in a revised manuscript

We therefore ask you to modify the manuscript according to the review recommendations. Your revisions should address the specific points made by each reviewer. Specifically, it would be useful to increase the number of simulations as suggested by reviewer 3 for figure 2. 

[LINK]

Yours sincerely,

Scott M. Williams

Section Editor: Natural Variation

PLOS Genetics

Hua Tang

Section Editor: Natural Variation

PLOS Genetics

Reviewer's Responses to Questions

**Comments to the Authors:**

Reviewer #1: Thank you to the authors for their thorough response - I have no further comments.

Reviewer #2: The authors have made a great effort to address my comments and improved their manuscript.

I have no further comments.

Reviewer #3: The authors have done a commendable job in addressing my concerns, and in my opinion have produced a very interesting piece of work. I would also like to thank them for formatting their revision so clearly and thus making it very easy to follow the changes in the revision process.

I only have two further comments:

1) Figure 2 shows some pretty dramatic non-monotonicity in the mean H_R(1) and H_R(2) statistics in the simulated samples. This highlights a point that (as discussed by the authors with reference to real data) the effect size may be relatively subtle and somewhat noisy. Would it be possible to increase the sample size/number of simulations performed for Figure 2, so that the nonmonotonicity is ultimately averaged out, giving the reader a sense of what can be expected under ideal conditions. It is also good to see the effect of limited sampling (and I very much like how the authors accomplished to show both in Figure 6). I don’t know whether this is computationally feasible at this point, but if at all possible, it would be very scientifically informative.

2) Lines 279, 280: “Our simulations predict that after deleterious variants increase above a  frequency of 6% they will no longer have lower D’ values than neutral alleles.” — I recommend including a brief clause along the lines of “under the assumed simulation parameters” just to emphasize to the reader that this is not an absolute statement about a definitive cutoff in human genomes.

**Have all data underlying the figures and results presented in the manuscript been provided?**

Reviewer #1: Yes

Reviewer #2: Yes

Reviewer #3: Yes

PLOS authors have the option to publish the peer review history of their article (what does this mean?). If published, this will include your full peer review and any attached files.

Reviewer #1: No

Reviewer #2: No

Reviewer #3: No

---

## [Editor Report · Decision Letter 2]

22 Jun 2021

Dear Dr Lohmueller,

We are pleased to inform you that your manuscript entitled "Negative linkage disequilibrium between amino acid changing variants reveals

interference among deleterious mutations in the human genome" has been editorially accepted for publication in PLOS Genetics. Congratulations!

Yours sincerely,

Scott M. Williams

Section Editor: Natural Variation

PLOS Genetics

Hua Tang

Section Editor: Natural Variation

PLOS Genetics

Comments from the reviewers (if applicable):

**Data Deposition**

http://datadryad.org/submit?journalID=pgenetics&manu=PGENETICS-D-20-00326R2

**Press Queries**

---

## [Editor Report · Acceptance letter]

23 Jul 2021

PGENETICS-D-20-00326R2 

Negative linkage disequilibrium between amino acid changing variants reveals
interference among deleterious mutations in the human genome 

Dear Dr Lohmueller, 

We are pleased to inform you that your manuscript entitled "Negative linkage disequilibrium between amino acid changing variants reveals
interference among deleterious mutations in the human genome" has been formally accepted for publication in PLOS Genetics! Your manuscript is now with our production department and you will be notified of the publication date in due course.

With kind regards,

Agota Szep

PLOS Genetics

On behalf of:
